# A 'Candidatus Liberibacter asiaticus' effector SDE2470 facilitates citrus transcription factor CsVOZ2 degradation via BRUTUS E3 ligases

Shimin Fu[1,2]*, Xiaofeng Yang[1,2], Haoqing Zhao[1,2], Zuhui Yang[1], Jiajun Wang[1,2], Mingyue Qin[1,2], Changyong Zhou[1,2], Xuefeng Wang[1,2]*

1 Integrative Science Center of Germplasm Creation in Western China Science City, Citrus Research Institute, Southwest University, Chongqing, China, 2 National Citrus Engineering Technology Center, Citrus Research Institute, Southwest University, Chongqing, China

* minniefu@swu.edu.cn (SF); xfwang@swu.edu.cn (XW)

## Abstract

Citrus Huanglongbing (HLB), a devastating disease caused by the Gram-negative bacterium 'Candidatus Liberibacter asiaticus' (CLas), poses serious threats to global citrus production and lacks effective control strategies. Previously, SDE2470 (CLIBASIA_02470) was identified as a Sec-dependent effector that contributes to CLas pathogenesis, although its underlying molecular mechanisms were not fully elucidated. In this study, SDE2470 was found to target a citrus vascular one-zinc-finger transcription factor CsVOZ2. CsVOZ2 overexpression (CsVOZ2-OE) in transgenic citrus plants significantly suppressed CLas colonization, whereas its RNA interference (RNAi) in citrus hairy roots enhanced susceptibility to CLas. Additionally, CsVOZ2-OE significantly increased reactive oxygen species (ROS) and abscisic acid (ABA) contents accumulation and activated related genes expression. Further investigation revealed that the E3 ligase CsBTS1 directly interacts with CsVOZ2 and promotes its degradation via the 26S proteasome pathway. CsBTS1E3-OE in citrus hairy roots markedly enhanced CLas proliferation. Importantly, SDE2470 directly interacts with CsBTS1E3 and strengthen CsBTS1E3-CsVOZ2 interaction. Meanwhile, SDE2470 strengthened the E3 ligase activity of CsBTS1, promoting CsVOZ2 degradation. Taken together, these findings support a model in which SDE2470 hijacks CsBTS1 to destabilize CsVOZ2, thereby disrupting ROS- and ABA-dependent immunity and promoting CLas infection in citrus.

## Author summary

Huanglongbing (HLB), a devastating citrus disease, is caused by the bacterium *Candidatus* Liberibacter asiaticus (CLas). This study demonstrates that the citrus transcription factor CsVOZ2 confers resistance to CLas, with overexpression of

**Data availability statement:** All relevant data are within the paper and its Supporting Information files.

**Funding:** This work was financially supported by the National Natural Sciences Foundation of China (32402312 to SMF and U23A20196 to XFW; https://www.nsfc.gov.cn/), the Natural Sciences Foundation of Chongqing, China (CSTB2024NSCQ-MSX1133 to SMF; https://kjj.cq.gov.cn/) and the Special Fund for Youth Team of Southwest University (SWU-XJLJ202310 to XFW; https://www.swu.edu.cn/). The funders had no role in study design, data collection and analysis, decision to publish, or preparation of the manuscript.

**Competing interests:** The authors have declared that no competing interests exist.

CsVOZ2 enhances immunity, whereas its knockdown increases susceptibility. However, the *C*Las effector SDE2470 hijacks CsVOZ2 and promotes its degradation, thereby disrupting ROS- and ABA-mediated immune responses. These findings provide new insights into *C*Las pathogenicity and host interactions, and identify promising targets for engineering HLB-resistant citrus varieties.

## Introduction

Citrus Huanglongbing (HLB), also known as citrus greening disease, is one of the most devastating diseases affecting citrus production worldwide. The disease is caused by the phloem-limited bacterium 'Candidatus Liberibacter asiaticus' (*C*Las) [1], which is transmitted by the Asian citrus psyllid (*Diaphorina citri*). HLB is characterized by symptoms such as yellowing of leaves, stunted growth, and bitter, misshapen fruits, ultimately leading to tree decline and significant economic losses [2]. Despite extensive research, effective control measures for HLB remain elusive, largely due to the difficulty to culture *C*Las *in vitro* and the complexity of its interactions with host plants.

Although *C*Las genome is highly compact and lacks both type III (T3SS) and type IV secretion systems (T4SS), it nevertheless harbors a complete general secretion (Sec) pathway [3,4]; and some Sec-dependent effector (SDEs)-mediated pathogeneses have been revealed. SDE1 (CLIBASIA_05315) was identified to trigger hypersensitive response (HR)-based cell death and suppresses plant immunity by targeting citrus papain-like cysteine proteases (PLCPs) (Marco et al., 2016, 2018). Similarly, SDE15 (CLIBASIA_04025) inhibits HR and suppresses defense responses by interacting with citrus accelerated cell death protein ACD2 [5]. SDE4405 (CLIBASIA_04405) and SDE3 (CLIBASIA_00420) contributes to *C*Las invasion and HLB progression in manipulation of citrus autophagy by targeting autophagy related protein 8 (ATG8s) and cytosolic glyceraldehyde-3-phosphate dehydrogenases GAPCs, respectively [6]. AGH17470 and AGH17488 were identified to induce strong immune responses and inhibit APX6 enzymatic activity, respectively [7,8]. Another effector, RNaseHI delays flowering via interacting with a citrus B-Box zinc finger protein [9]. Furthermore, SDE19 (CLIBASIA_05320) perturbed vesicle trafficking related defenses [10]. Notably, SDE5 (CLIBASIA_02470, designated as SDE2470 in this study), exhibiting bacterial C-type lysozyme inhibition activity, employs a dual strategy: enhancing PUB21-mediated degradation of MYC2 via the 26S proteasome system (UPS) and disrupting MYC2 dimerization; thereby suppressing jasmonate (JA)-mediated citrus immunity against *C*Las and impairs terpene-based anti-herbivore defenses [11,12].

The UPS, a highly conserved pathway responsible for protein degradation and regulation, plays a central role in plant immunity by controlling the stability of immune-related proteins, such as resistance (R) proteins, transcription factors, and signaling components. E3 ubiquitin ligases, which determine substrate specificity in the ubiquitination process, are frequent targets of pathogen effectors [13]. Effectors can directly target E3 ligase, affecting its activity and its-mediated substrates' stability, thereby facilitating pathogen colonization and disease progression [14,15]. Iron sensor BRUTUS (BTS)

is a plant-specific iron-binding E3 ubiquitin ligase that plays a critical and conserved role in the maintenance of iron homeostasis across various plant species [16,17]. It contains two to three iron-binding hemerythrin (Hr) domains and a RING-type zinc finger domain (34), enabling it to sense cellular iron status and mediate the ubiquitination and subsequent degradation of key regulators involved in iron uptake and regulation. *Arabidopsis* BTS and BTS-like functions in iron homeostasis regulation by interacting with IRON MAN and PYE-like (PYEL) transcription factors via RING domain and mediated their degradation through 26S proteasome-mediated pathways [16,18]. *Arabidopsis* BTS also promotes the nucleus translocation and degradation of VOZ1/2 (Vascular plant one-zinc finger) to against drought [19]. Intriguingly, *Arabidopsis* BTS was hijacked by effector AvrRps4 to facilitate iron acquisition and *Pst* DC 3000 colonization [20]. Therefore, BTS and BTS-like proteins possess specific features that are unique to the plant kingdom and enable them to perform multiple biological functions.

The VOZ transcription factors constitute a class of regulatory proteins integral to plant growth, development and stress responses. In Arabidopsis, AtVOZ2 regulates appropriate flowering and seed germination [21–23], and in soybean, GmVOZ1A regulates oil synthesis [24]. Moreover, numerous studies have elucidated their pivotal role in plant disease resistance. Notably, rice *OsVOZ1/2* double-silenced plants display remarkable reduction of reactive oxygen species (ROS)-dependent cell death responses and resistance to the *Magnaporthe oryzae* in the *Piz-t* background [25]. In peach fruit, it has been evidenced that VOZ-dependent priming of salicylic acid-dependent defense against *Rhizopus stolonifer* [26]. Arabidopsis AtVOZ1/2 paired with β-aminobutyric acid receptor AtIBI1 to regulate abscisic acid (ABA) signaling and callose associated defense [27], and potato StVOZ1/2 interacted with StIBI1 finely regulate potato resistance to late blight [28]. Emerging evidence indicates that the stability of VOZ transcription factors is tightly regulated by the ubiquitin-mediated degradation [19]. Nerveless, its roles in citrus immunity has not been clarified.

These studies have provided valuable insights into the mechanisms underlying *C*Las-citrus interactions, but the full scope of these interactions remains largely unexplored. Previously, *C*Las effector SDE2470 has been identified to facilitate *C*Las infection [12], but its pathogenesis remains poorly characterized. In this study, we revealed that SDE2470 hijacks citrus vascular transcription factor CsVOZ2 for ubiquitin-mediated degradation through enhancing the ligase activity of CsBTS1, thereby comprising plant immunity against *C*Las infection. The results provide new insight into our understanding of *C*Las-citrus interactions.

## Results

### SDE2470 served as a broad-spectrum suppressor of basal immunity to facilitate pathogen infection

Effector SDE2470 (CLIBASIA_02470) was identified as a virulence factor promoting *C*Las infection in citrus [12], though its role in basal immunity remained undefined. Therefore, its function on basal immunity was tested via PVX-mediated transient expression. Transient expression of PVX-SDE2470 significantly attenuated BAX- and INF1-induced reactive oxygen species (ROS) burst in *Nicotiana benthamiana* leaves compared to GFP or buffer controls via 3,3'-diaminobenzidine (DAB) staining and quantification analysis (Fig 1A-1B). Further, to understand whether SDE2470 associates other pathogen infection, *Pseudomonas syringae* pv. *tomato* DC3000 (*Pst* DC3000) and *Xanthomonas citri* subsp. *citri* (*Xcc*) were tested in *SDE2470*-overexpressing transgenic *Arabidopsis* and citrus plants (OE #4), respectively. *SDE2470*-OE *Arabidopsis* and citrus plants were verified with GUS-staining, PCR and RT-qPCR assays (S1 Fig). With *Pst* DC3000 and *Xcc* inoculation, both *SDE2470*-OE transgenic Arabidopsis and citrus leaves displayed exacerbated disease symptoms *vs.* wild-type (WT) (Fig 1C and 1E). The observation was further supported with greater *Pst* DC3000 and *Xcc* biomass (Fig 1D and 1F-1G). Therefore, these results support that SDE2470 served as a board-spectrum suppressor of basal immunity to facilitate pathogen infection.

### SDE2470 physically interacts with *C. sinensis* vascular transcription factor CsVOZ2

To elucidate the mechanism by which SDE2470 suppresses host immunity, we performed a yeast two-hybrid (Y2H) screen using SDE2470 as bait against a *Citrus sinensis* cDNA library, and a vascular plant one-zinc finger transcription

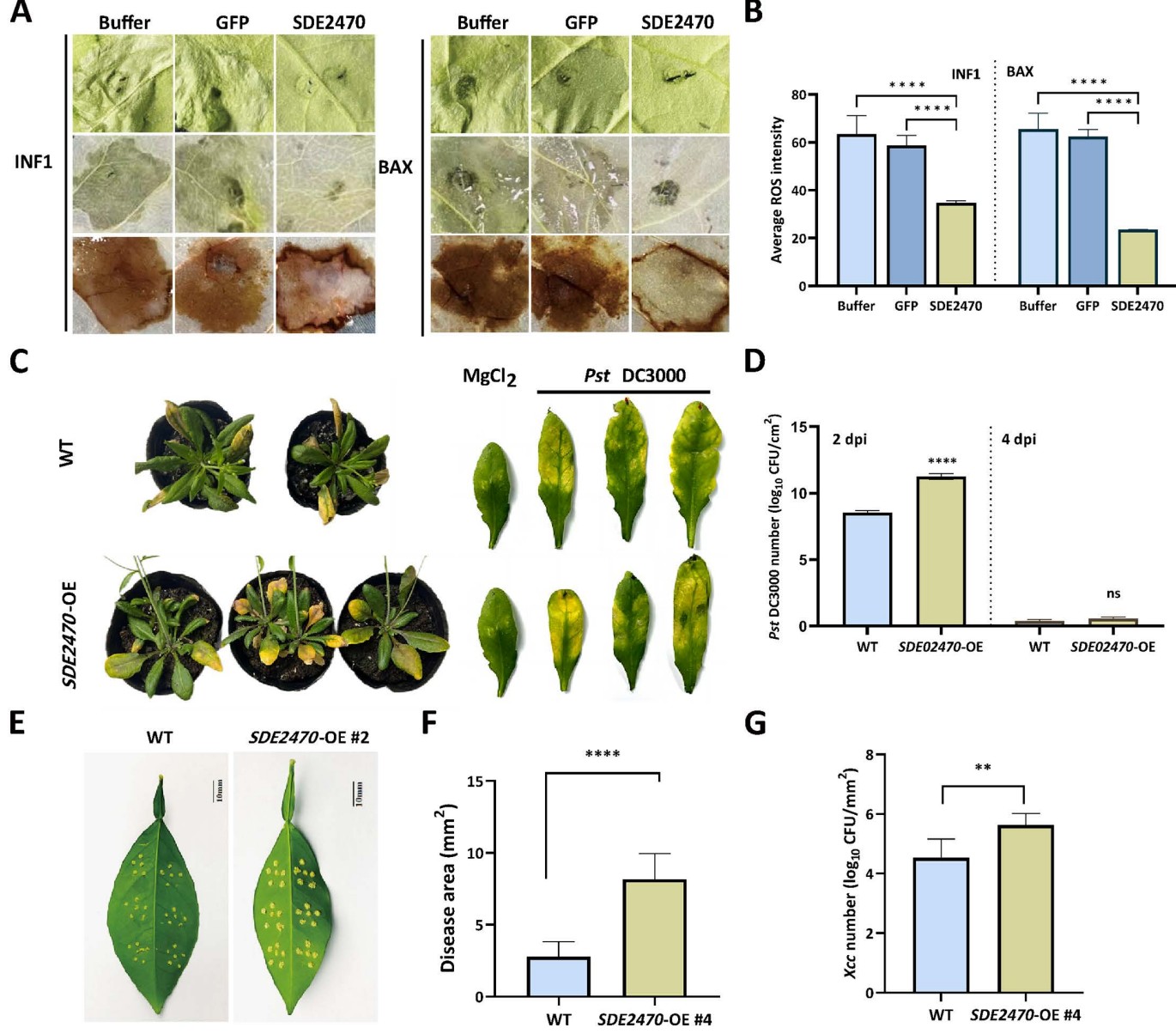

**Fig 1. Biological function analyses of effector SDE2470 against *Pst* DC3000 and *Xcc*.** (A) Suppression of BAX- and INF1-induced cell death and ROS burst by SDE2470 in *Nicotiana benthamiana*. Leaves were pre-infiltrated with PVX-SDE2470, PVX-GFP (negative control), or buffer 24 hours prior to PVX-BAX or PVX-INF1 infiltration. Symptom development was monitored, followed by ethanol decolorization and DAB staining for ROS observation. (B) Quantitative analysis of ROS intensity shown in (A) using ImageJ. The experiment was repeated twice and with four biological replicates each time. Statistical significance were analyzed by two-way ANOVA test (****$P < 0.0001$). (C-D) Symptom observation and bacterial biomass assays on SDE2470-OE *Arabidopsis thaliana* leaves against *Pst* DC3000 challenge. The evaluation was done with three transgenic lines with wild-type (WT) served as controls. Statistical significance was determined using two-way ANOVA test (****$P < 0.0001$). (E-G) Susceptibility assay of *SDE2470*-OE transgenic *Citrus sinensis* leaves against *Xcc* infection. Five leaves from *SDE2470*-OE #4 transgenic lines were inoculated with *Xcc* suspension and photographed at 15 days post-inoculation (dpi). Wild-type (WT) citrus leaves served as controls. Scale bar = 10 mm. Subsequently, colony size and bacterial biomass were assays. Statistical significance of both colony size and bacterial biomass were determined via Student's *t*-test (**$P < 0.01$, ****$P < 0.0001$).

factors CsVOZ2 (Cs2g16020) was identified. It belongs to the NAC superfamily and shares 55–61% amino acid sequence similarity with *Arabidopsis thaliana* VOZ1/2 (S2A Fig). Meanwhile, *CsVOZ2* was highly upregulated in response to *C*Las infection (S2B Fig). Then, the SDE2470-CsVOZ2 interaction were validated using luciferase complementation assays (LCA) (Fig 2A). BiFC assay further revealed the SDE2470 interacts with CsVOZ2 in the nucleus and cytoplasm (Fig 2B). Furthermore, when Flag-CsVOZ2 co-expressed with myc-SDE2470 or myc-GUS, it was co-immunoprecipitated by myc-SDE2470 rather than myc-GUS (Fig 2C). GST-pulldown assay demonstrated that CsVOZ2-His was specifically pulled down by GST-SDE2470 but not by GST (Fig 2D). Given that AtVOZs have been shown to function as dimers [29], we investigated whether CsVOZ2 could form homodimers. LCA assays confirmed the self-interaction of CsVOZ2 (S3A Fig), and BiFC assays indicated that these homodimers form predominantly in the nucleus (S3B Fig). Moreover,

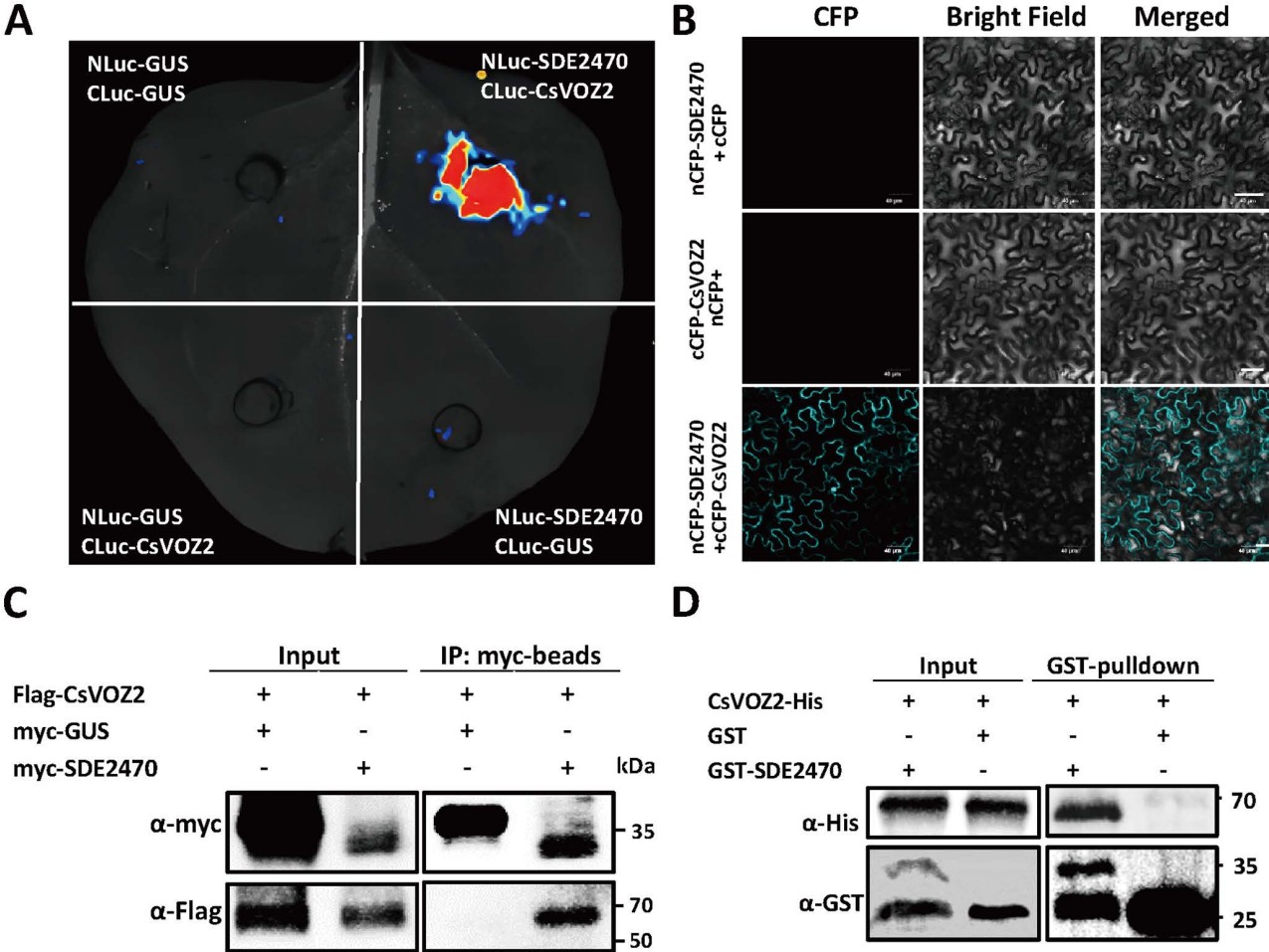

**Fig 2. Verification of the interaction between SDE2470 and CsVOZ2.** (A-B) Protein-protein interaction analysis using luciferase complementation assay (LCA) and bimolecular fluorescence complementation (BiFC). SDE2470 and CsVOZ2 were cloned into LCA vectors (NLuc and CLuc) and BiFC vectors (pCV-nCFP and pCV-cCFP), respectively. NLuc-GUS and CLuc-GUS constructs served as negative controls. All constructs were transformed into *Agrobacterium tumefaciens* strain GV3101 and transiently co-expressed in *N. benthamiana* leaves. Fluorescence signals were observed at 2 days post-infiltration (dpi). Scale bars = 40 μm. (C) Co-IP verification of SDE2470-CsVOZ2 interaction via transient expression in *N. benthamiana* leaves. (D) *In vitro* interaction validation by GST-pulldown assay. GST-tagged SDE2470 and GST control proteins were expressed in *Escherichia coli*, purified using GST antibody coupled beads, and incubated with *E. coli* lysates containing His-tagged CsVOZ2. Bound proteins were eluted and detected by immuno-blotting using anti-GST and anti-His antibodies. All the experiments were repeated at least three times.

co-transformation of BD-CsVOZ2 and AD-empty vector enabled yeast growth (S3C Fig), indicating its transcriptional activation activity.

To understand the subcellular localization of their interactions, SDE2470 and CsVOZ2 were fused at the N-terminal of RFP and GFP, respectively, and transiently expressed in *N. benthamiana* leaves. When expressed individually, SDE2470 localized to multiple subcellular compartments, including the nucleus and cytoplasm (S4A Fig), consistent with previous reports [30]. Similarly, CsVOZ2 exhibited dual localization in the nucleus and cytoplasm (S4A Fig). Co-expression of SDE2470 with CsVOZ2 resulted in fluorescence signals in both compartments (S4B Fig), aligning with the interaction compartments observed via BiFC. Taken together, the findings demonstrated that SDE2470 interacts with CsVOZ2 in both the nucleus and cytoplasm.

**CsVOZ2 enhances citrus immunity against *C*Las colonization**

To characterize the biological role of CsVOZ2 in the process of *C*Las infection, two *CsVOZ2*-OE transgenic lines (#1 and #2) and three RNAi hairy root lines (#B1, #B8 and #B9) were generated, verified with fluorescence observation, PCR, RT-qPCR and WB assays (S5A-E Fig), and graft-inoculated with *C*Las-infected buds (Fig 3). At one month post inoculation (1 mpi), no *C*Las was not detected either in *CsVOZ2*-OE nor WT plants. By 2 and 3 mpi, *C*Las was detectable in both groups, but its population was significantly lower in *CsVOZ2*-OE than in WT control (Fig 3B). Conversely, *C*Las colonized more rapidly in *CsVOZ2*-RNAi *vs.* EV citrus hairy roots (Fig 3C). As HLB progressed, WT plants began to exhibit characteristic stunted and chlorotic symptoms in newly emerged leaves at 3 mpi, whereas *CsVOZ2*-OE plants remained largely asymptomatic (Fig 3A).

To investigate how CsVOZ2 enhances citrus immunity, RNA-seq was performed with healthy *CsVOZ2*-OE (line #2) citrus leaves. Overall, a total of 3579 differentially expressed genes (DEGs) were identified in comparison to healthy WT, with more DEGs upregulated (2264 genes) than downregulated (1315 genes) (S1 Table). These DEGs were highly enriched in protein processing, plant-pathogen interaction, cell wall modification, carbohydrate metabolism, MAPK signaling, and hormone signal transduction pathways (S6A–S6B Fig). More specifically, the overexpression of *CsVOZ2* triggers cell-surface receptors (such as FLS2) and RBOHD, alongside calcium-signaling components (including CNGCs, CDPKs, and CMLs). This initial signaling leads to a burst of reactive oxygen species (ROS) and the accumulation of peroxidases. Subsequently, MAPK cascades are activated, which in turn probably phosphorylate and activate a suite of transcription factors, such as WRKY, MYB/C, bZIP, and TCP types. These transcription factors then orchestrate downstream defenses by reprogramming hormone signaling and secondary metabolism, as well as inducing the expression of Pathogenesis-Related (PR) proteins. In parallel, the expression of genes involved in wax and cutin biosynthesis (e.g., CER1, PXG, FAR, FACR) is upregulated to reinforce the cell wall-based physical barriers. However, the ubiquitin-mediated proteolysis system is also activated to degrade accumulated proteins to maintain cellular homeostasis (S6C Fig). Furthermore, tissue-specific expression alignment identified 222 DEGs enriched in the phloem complex, including phloem parenchyma, companion cells, sieve elements, pole pericycle, and phloem/pericycle tissues (S6D Fig and S2 Table). Functional annotation indicated their involvement in protein folding, degradation, signaling, as well as carbohydrate, starch, and sucrose metabolism (S6E Fig). Therefore, RNA-seq profiling indicates that CsVOZ2 triggers multi-layered defense responses.

To confirm the RNA-seq data and regulation roles of CsVOZ2 in *C*Las invasion, 15 DEGs and marker genes (including *CsRbohD*, *CsFLS2*, *CsAPX2*, *CsM2K6*, *CsTGA2*, *CsWRKY229/33*, *CsPAD4*, *CsEDS1*, *CsNPR1*, *CsPR5*, *CsABI5*, *CsKCS20*, *CsGASA6* and *CsSTP13*) associated with ROS, ABA and JA/SA pathways were assayed via RT-qPCR. As expected, these genes showed overall upregulation in healthy *CsVOZ2*-OE plants, whereas consistently down-regulated in healthy RNAi hairy roots (Fig 3D). Given previous reports linking VOZ proteins to ROS and ABA pathways [25,31], ROS and ABA contents were measured and showed significant accumulation in healthy *CsVOZ2*-OE (#1 and #2) *vs.* WT plants (Fig 3E-3F). Collectively, these findings strongly support that CsVOZ2 enhances citrus immunity against *C*Las colonization via the activation of ROS- and ABA-mediated immunity.

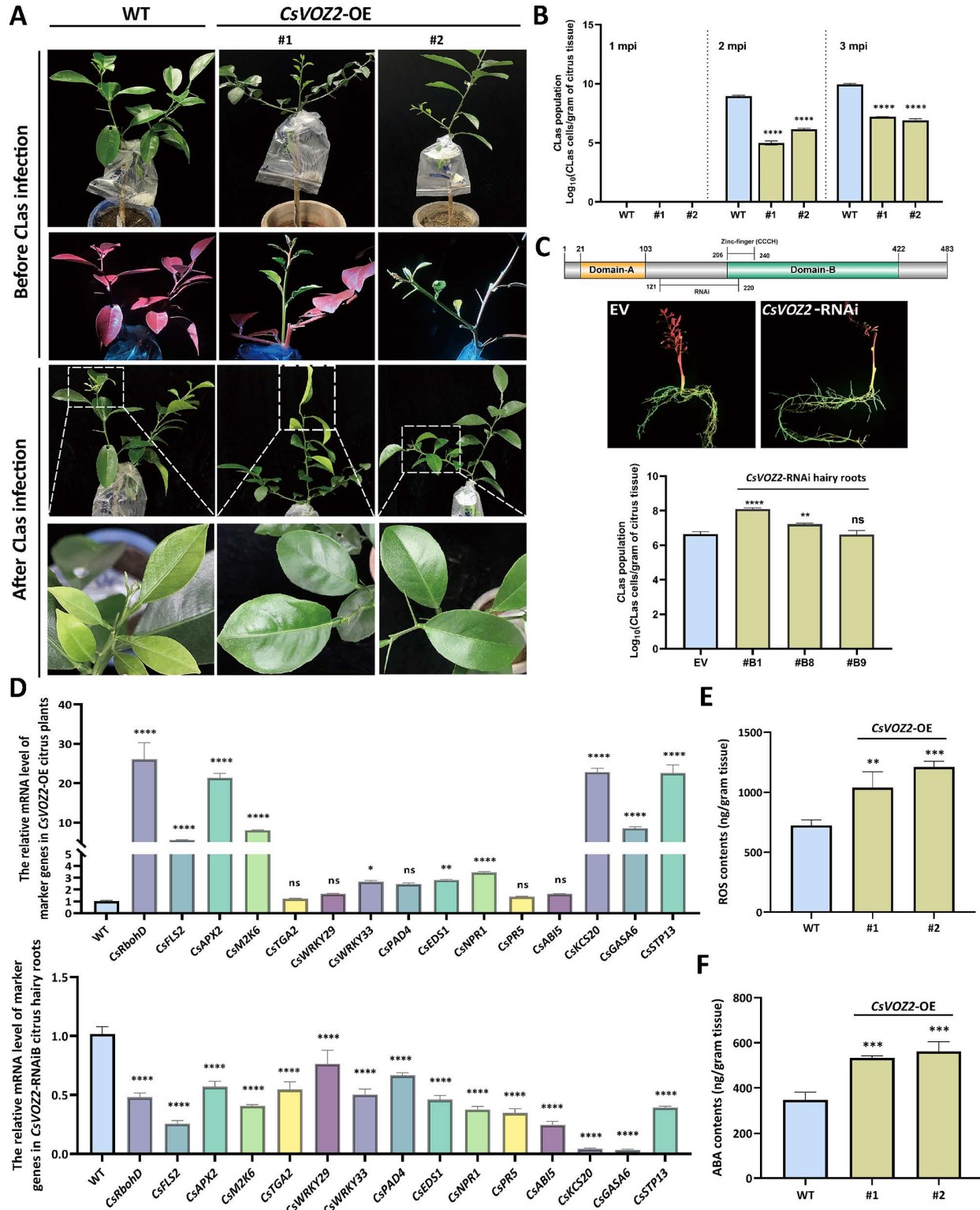

**Fig 3. Biological function evaluation of *CsVOZ2* transgenic citrus plants/hairy roots in response to 'Candidatus Liberibacter asiaticus' (CLas) challenge.** (A) *CsVOZ2*-OE transgenic citrus plants verification with GFP fluorescence observation and symptom development at 3 months post CLas inoculation (mpi). Two *CsVOZ2*-OE transgenic lines with three replicates of each line were graft-inoculated with CLas-infected buds at the base where

mock-/transgenic-branches grafted. Then the grafted buds were sealed with plastic bag with wet paper towels to maintain moisture. (B) CLas population assessment in citrus plants. Equal weight of midribs from 2-3 newly emerged and mature leaves exhibiting suspected symptoms were pooled for CLas assays. Detection was conducted monthly, starting at 1 mpi and continuing until 3 mpi. CLas population was assayed by qPCR and analyzed with two-way ANOVA test (****$P < 0.0001$). (C) CLas population assessment in *CsVOZ2*-RNAi transgenic citrus hairy roots. Schematic diagram of CsVOZ2 protein was shown with interference region marked. Three independent RNAi hairy root lines were verified with GFP fluorescence observation and grafted inoculated with CLas-infected buds on the aerial part. Then its population was assayed by qPCR at 1 mpi and analyzed with two-way ANOVA test (ns indicates no significant difference, **$P < 0.01$, ****$P < 0.0001$). (D) The expression levels of plant immunity marker genes and differentially expressed genes identified from transcriptome profiling were assayed via RT-qPCR in healthy *CsVOZ2*-OE citrus plants and RNAi hairy roots. Two independent OE and three RNAi lines were analyzed, with transcript levels normalized to citrus *actin 7*. The relative gene expression was calculated and averaged across biological replicates. Statistical significance was determined using one-way ANOVA test (*$P < 0.05$, **$P < 0.01$ and ****$P < 0.0001$). (E-F) ROS and ABA contents were measured in leaves of two independent *CsVOZ2*-OE citrus lines. For each transgenic line, three individual leaves were assayed as biological replicates. Statistical significance was assessed by one-way ANOVA test (**$P < 0.01$, ***$P < 0.001$).

## CsBTS1E3 directly interacts with CsVOZ2 and mediates its degradation

While the biological function of CsVOZ2 has been characterized, how SDE2470 affects its function remains elusive. To elucidate this molecular mechanism, we firstly examined the transcriptional regulation of CsVOZ2 by SDE2470. Quantitative RT-qPCR assay revealed no significant change in *CsVOZ2* mRNA level in *SDE2470*-OE *vs*. WT controls (S7A Fig). Then we hypothesized that SDE2470 might regulate CsVOZ2 protein stability. Thereupon myc-tagged SDE2470 and flag-tagged CsVOZ2 constructs were transiently co-expressed in *N. benthamiana* leaves. Western blot assay demonstrated that the presence of SDE2470 significantly reduced CsVOZ2 protein accumulation compared to the control, whereas the degradation effect was substantially attenuated by treatment with 26S proteasome inhibitor of MG132 (Fig 4A).

Then we speculated that some host E3 ligases were involved in CsVOZ2 degradation. Therefore, we performed RNA-seq on *SDE2470*-OE transgenic citrus plants. The transcriptomic profile revealed the upregulation of several E3 ligases genes, including BTS (containing a RING domain), as well as RHA, RHY, and PUBs (containing U-box domain) (S7B Fig). The transcriptional upregulation of CsBTS1 (Cs2g12820) but not CsBTS2 (Cs7g27090) was further confirmed by RT-qPCR assay (S7C Fig). From NCBI database, three isolates were retrieved for CsBTS1 and the longest one (XP_006468730.1) features three hemerythrin domains (HHEs), one zf-CHY domain, and a RING-H2 domain (S7D Fig). Despite numerous attempts, we were unable to clone the full-length of CsBTS1 CDS. Consequently, a truncate containing the E3 RING domain (residues 782–1178 of the full length) was successfully cloned and fused to the N terminal of GFP. Subcellular localization analysis showed that the CsBTS1E3-GFP exclusively distributed in the nucleus (S7E Fig). Then the truncate was utilized for subsequent studies.

We next investigated whether CsBTS1E3 directly interacts with CsVOZ2. LCA assays suggested that CsBTS1E3 interacts with CsVOZ2 (Fig 4B). BiFC further demonstrated their interactions occurring in the nucleus (Fig 4C). Their interaction was further validated with GST-pulldown assay *in vitro* (Fig 4D). Subsequently, whether CsBTS1E3 possess E3 ligase activity was tested in *vitro* ubiquitination assay. Since histidine (H) residues within the conserved RING domain are known to be critical for E3 activity [32], we generated a mutant CsBTS1E3M by substituting two key residues (H358Y/H361Y). In the presence of E1/E2/Ubi-Flag, CsBTS1E3 underwent robust self-ubiquitination, whereas the CsBTS1E3M mutant lost this activity (S7F Fig). We then examined whether CsBTS1E3 mediates the degradation of CsVOZ2 in planta. Myc-tagged CsBTS1E3 or CsBTS1E3M was co-expressed with Flag-tagged CsVOZ2 in *N. benthamiana* leaves, CsBTS1E3 promoted the degradation of CsVOZ2 in comparison to CsBTS1E3M. This degradation was reversed upon treatment with the proteasome inhibitor MG132 (Fig 4E). Consistently, in an *in vitro* ubiquitination assay, only CsBTS1E3 could mediate the ubiquitination of CsVOZ2 rather than its mutant CsBTS1E3M (Fig 4F). Hence, these results demonstrate that CsBTS1E3 directly interacts with CsVOZ2 and promotes its degradation via the 26S proteasome pathway.

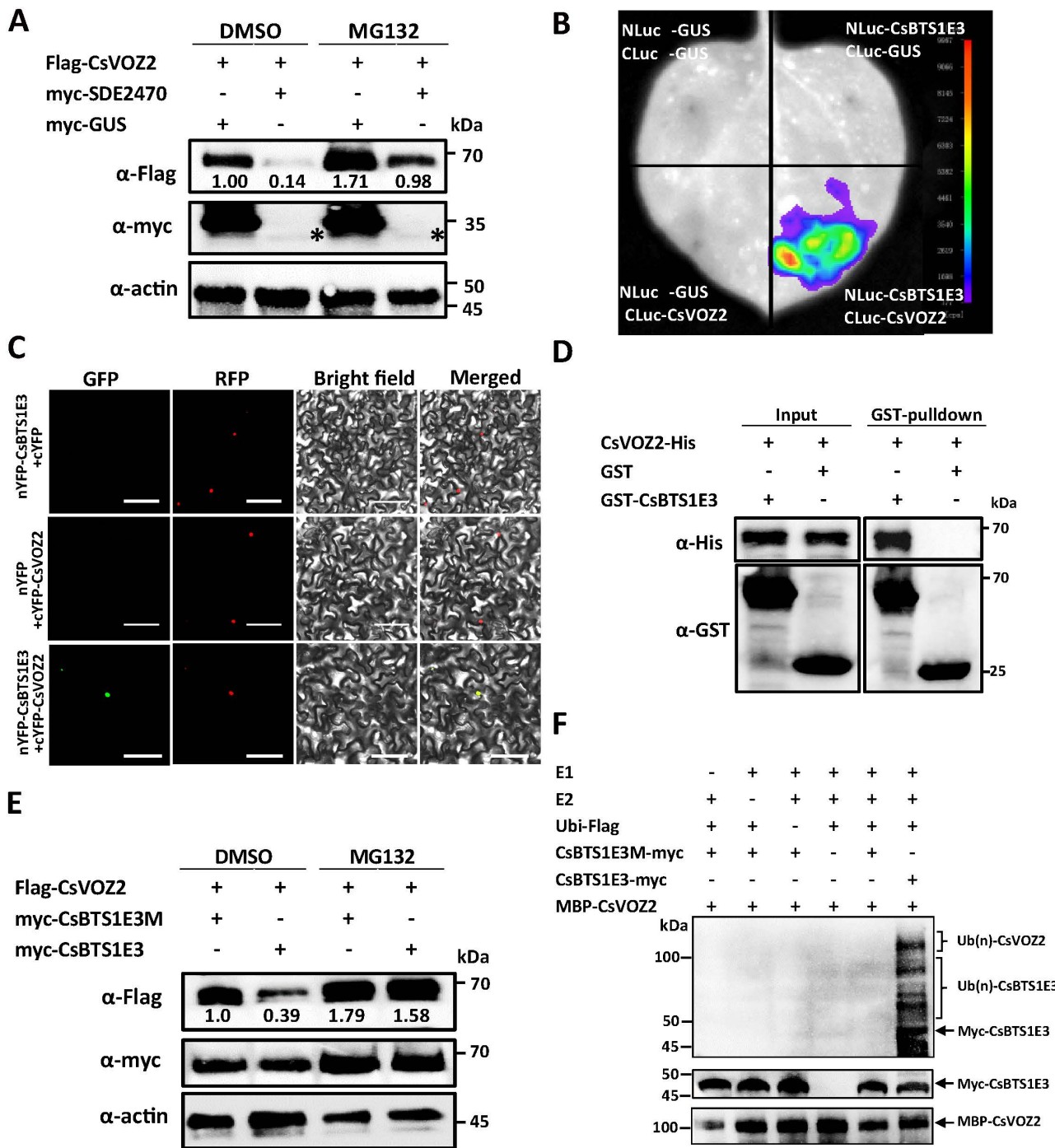

**Fig 4. Regulatory effects of SDE2470 on CsVOZ2 protein stability.** (A) Assessment of SDE2470 effect on CsVOZ2 protein stability. Myc-tagged SDE2470 and flag-tagged CsVOZ2 were transiently co-expressed in *N. benthamiana* leaves and treated with 26S proteasome inhibitor MG132 (50 μM), with myc-GUS expression and DMSO treatment as controls. Total protein was extracted at 2 dpi for immunoblot analysis using anti-flag and anti-myc antibodies with actin served as an internal control. The abundance of Flag-CsVOZ2 protein in each lane was quantified using ImageJ and normalized against the corresponding actin abundance. Then the ratio obtained from the first lane was set to 1.00 for relative comparison. The asterisk indicates the faint band corresponding to myc-SDE2470. (B-D) CsVOZ2-CsBTS1E3 interaction was verified with LCA, BiFC and GST-pulldown assays. In LCA, CLuc-GUS and NLuc-GUS were used as controls. In BiFC, H2B-RFP was used nuclear localization marker. In GST-Pulldown, GST was included as a control. (E) Assessment of CsBTS1E3 effect on CsVOZ2 protein stability. Myc-tagged CsBTS1E3 or CsBTS1E3M was transiently expressed with flag-CsVOZ2

in *N. benthamiana* leaves with or without MG132 (50 µM) treatment, and DMSO served as control. Total protein was extracted at 2 dpi for immunoblot analysis using anti-flag and anti-myc antibodies. The abundance of flag-tagged CsVOZ2 protein in each lane was quantified using ImageJ and normalized against the corresponding actin abundance. Then the ratio of the relative CsVOZ2 abundance in the first lane was set to 1.00 for relative comparison. (F) The ubiquitination ability assay of CsBTS1E3 on CsVOZ2 protein *in vitro.* Myc-CsBTS1E3M with ligase activity sites mutated was served as comparison. All experiments were performed three times.

## CsBTS1E3 overexpression facilitates *C*Las colonization in citrus hairy roots

To investigate the biological role of *CsBTS1E3* during *C*Las infection, we generated myc-tagged *CsBTS1E3*- and *CsBTS1E3M*-OE hairy roots with *Poncirus trifoliate.* Successful transformation and transcript overexpression were confirmed by PCR and RT-qPCR, respectively (Fig 5A-5C). Then five independent transgenic lines per construct were graft-inoculated with *C*Las-infected buds. Subsequent quantitative assays at 1 mpi revealed that *C*Las colonization was significantly enhanced in both *CsBTS1E3*- and *CsBTS1E3M*-OE hairy roots compared to the EV control (Fig 5D). Notably, although CsBTS1E3M lacks E3 ligase activity, the *C*Las titers did not differ significantly between *CsBTS1E3*- and *CsBTS1E3M*-OE lines. This may be attributed to the more stable and higher transcript levels of CsBTS1E3M compared to CsBTS1E3 (Fig 5C). To assess the immune status, immune marker genes were assayed by RT-qPCR. All these genes including *CsRbohD*, *CsEDS1*, *CsNPR1*, *CsPR1* and *CsWRKY22/33* showed overall downregulation patterns (Fig 5E). Taken together, these results indicate that CsBTS1E3 negatively regulates citrus immunity to facilitate *C*Las proliferation.

## SDE2470 strengthens CsBTS1E3-CsVOZ2 interaction and CsBTS1E3 ligase activity to promote CsVOZ2 degradation

On the basis of the above results, whether SDE2470 directly interacts with CsBTS1E3 was tested and their interaction was preliminarily observed with LCA (Fig 6A). Then BiFC further verified their interaction in the cytoplasm and nucleus (Fig 6B), and GST-pulldown confirmed their interaction *in vitro* (Fig 6C). Notably, SDE2470 significantly enhanced CsBTS1E3-CsVOZ2 interaction in comparison to the GUS control (Fig 6D), and it was supported with fluorescence intensity (Fig 6E). The presence of GUS and SDE2470 proteins was confirmed with WB assay (Fig 6F).

Given that SDE2470 strengthened CsBTS1E3-CsVOZ2 interaction, whether it enhances CsBTS1E3-mediated CsVOZ2 degradation was assayed *in vivo*. HA-SDE2470-GFP and Flag-CsVOZ2 were transient co-expressed with myc-CsBTS1E3 or myc-CsBTS1E3M in *N. benthamiana* leaves. With HA-SDE2470-GFP expression, it enhanced CsVOZ2 degradation, which was partially reversed by MG132 treatment (Fig 7A). In the light of physical interaction of SDE2470-CsBTS1E3 (Fig 6), SDE2470 was speculated to enhance CsBTS1E3 activity. Indeed, the presence of GST-SDE2470 enhanced myc-CsBTS1E3 ubiquitin activity via *in vitro* assay (Fig 7B). Collectively, SDE2470 strengthens the physical interaction of CsBTS1E3-CsVOZ2, thereby facilitating CsVOZ2 degradation via the 26S proteasome pathway.

## Discussion

Ubiquitination serves as a pivotal regulatory mechanism in plant-pathogen interactions, functioning as a double-edged sword that pathogens exploit to subvert host immunity. In the rice-*Magnaporthe oryzae* pathosystem, E3 ligase APIP10 orchestrates blast resistance through ubiquitin-mediated degradation of transcription factors OsVOZ1/2 [25]. This paradigm extends to citrus Huanglongbing, where *C*Las secretes effectors targeting key immune pathways. Notably, effector SDE5 (SDE2470 in this study) suppresses JA-mediated defenses by hijacking the PUB21 to promote MYC2 degradation [12]. Critically, in our study SDE2470 was found to enhance CsBTS1-mediated CsVOZ2 degradation via 26S proteasome pathway. This coordinated manipulation demonstrates how *C*Las repurposes the host ubiquitination machinery to dismantle citrus immunity.

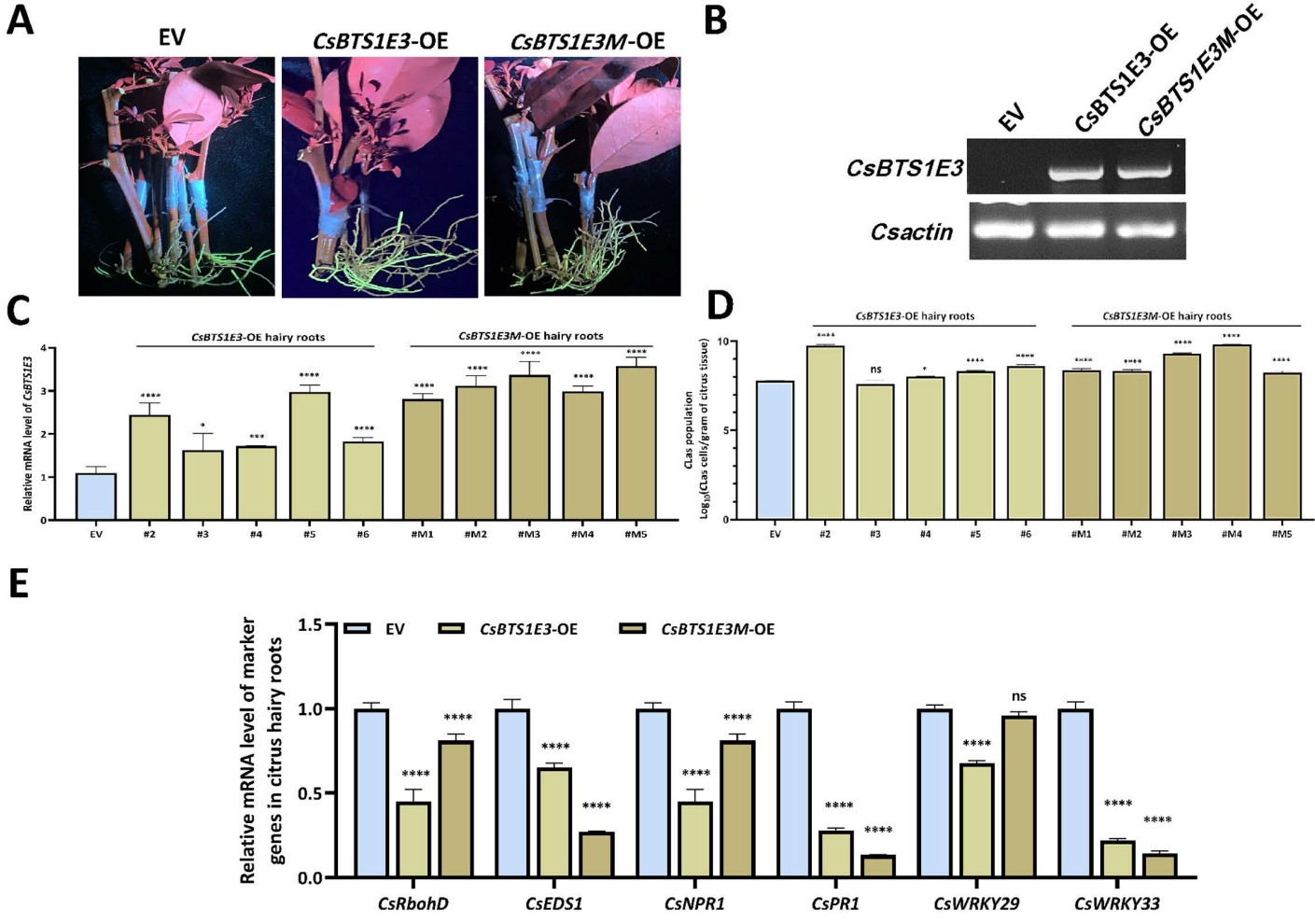

**Fig 5. Biological function evaluation of *CsBTS1E3*-OE via citrus hairy roots against *C*Las challenge.** (A-C) *CsBTS1E3*-OE and *CsBTS1E3M*-OE citrus hairy roots were validated with fluorescence observation, PCR by using vector-backbone and *CsBTS1E3*-specific primer pair and RT-qPCR assays. For gene expression analysis in (C), transcript levels were normalized to citrus actin 7, and statistical significance was determined by two-way ANOVA ('ns' indicates not significant. *$P < 0.05$, ***$P < 0.001$ and ****$P < 0.0001$). (D) *C*Las population assessment in *CsBTS1E3*-OE and *CsBTS1E3M*-OE citrus hairy roots. Five independent lines for each group were graft-inoculated with *C*Las-infected buds on the aerial parts, then hairy roots were subjected to *C*Las assays at 1 mpi by qPCR. Statistical significance was assessed by two-way ANOVA test ('ns' indicates not significant. *$P < 0.05$ and ****$P < 0.0001$). (E) The expression levels of plant marker genes were assayed via RT-qPCR in healthy *CsBTS1E3*-OE and *CsBTS1E3M*-OE citrus hairy roots. Three independent OE lines were analyzed as biological replicates. The relative gene expression was normalized to citrus actin 7. Statistical significance was determined using two-way ANOVA test (*$P < 0.05$ and ****$P < 0.0001$).

The susceptibility of VOZ transcription factors to pathogen manipulation is likely exacerbated by their dual attributes of strict spatial confinement and essential regulatory function. Although VOZ2 predominantly localizes in cytoplasm, it can be translocated from cytoplasm to nucleus with ABA treatment and E3 disease IBI1 presence (Yu et al., 2024, He et al., 2025). This compartmentalization mirrors *Magnaporthe oryzae*-induced nuclear translocation patterns observed, such as the nuclear shuttling of rice APIP5/OsbZIP24 [33] and BTS-mediated degradation of Arabidopsis VOZ proteins during drought stress [19]. The nuclear localization of CsVOZ2-effector interaction highlights a sophisticated pathogenic strategy to hijack host regulatory machinery. Further underscoring their vulnerability, VOZ proteins exhibit highly phloem-specific expression in Arabidopsis [23,29]. While single *voz* mutants in Arabidopsis show no overt developmental

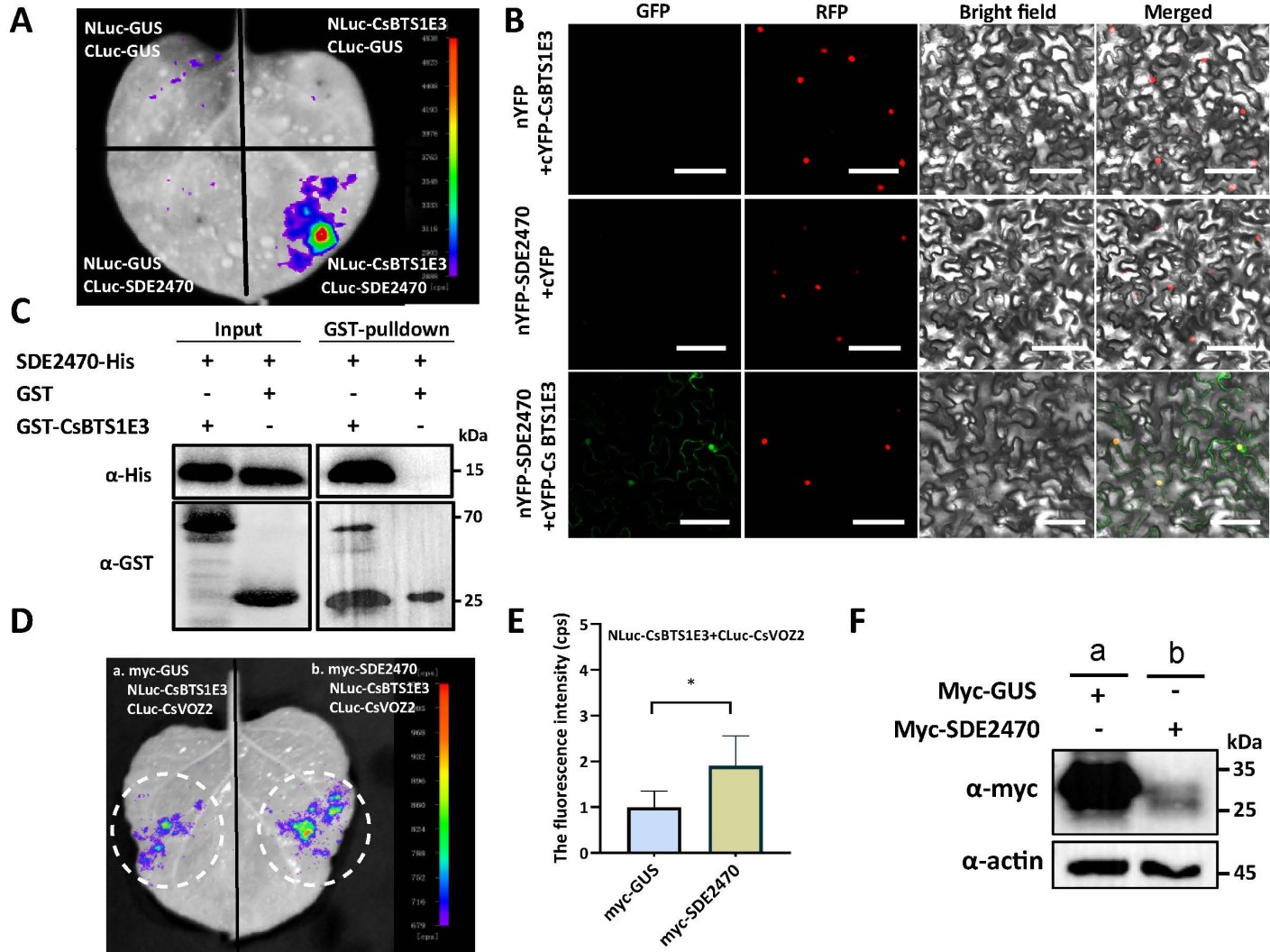

**Fig 6. SDE2470 interacts with CsBTS1E3 and its effect on CsBTS1E3-CsVOZ2 interaction.** (A-C) SDE2470-CsBTS1E3 interaction was confirmed via BiFC, LCA and GST-pulldown assays. In LCA assay, the paired constructs were transiently co-expressed in *N. benthamiana* leaves and the luciferase signal was observed with the supplementation of D-Luciferin substrate. In BiFC assay, the paired constructs were transiently co-expressed in *N. benthamiana* leaves with nuclear marker H2B and the fluorescence signal was observed 2 dpi with confocal microscope. GST-pulldown assay was conducted as described above. Scale bars = 50 μm. (D) The effect of SDE2470 on CsBTS1E3-CsVOZ2 interaction via LCA assay. The paried contructs NLuc-CsBTS1E3 and CLuc-CsVOZ2 were transiently co-expressed in *N. benthamiana* leaves with myc-SDE2470 serving as an effector. Myc-GUS was used as the control. (E) The luciferase intensity in region of interest (ROI) shown in (D) was quantified using ImageJ. Data were collected from five independent biological replicates, each consisting of paired leaf samples. Statistical significance between the treatment and control groups was determined by paired Student's *t*-test (*$P < 0.05$). (F) The presence of effector proteins SDE2470 and GUS in (D) was confirmed via WB assay. All experiments were conducted three times with comparable results.

defects, *voz1voz2* double mutants display severe phenotypes-including impaired phloem transport [34], indicating functional redundancy and a critical role in vascular function. This spatial restriction would create a strategic convergence with *C*Las's niche as an obligate phloem colonizer [35,36]. It is intriguing that a recent report demonstrated *OsVOZ2* knockout compromised shoot height but effectively strengthened defense against phloem-feeding herbivores [37]. Nevertheless,

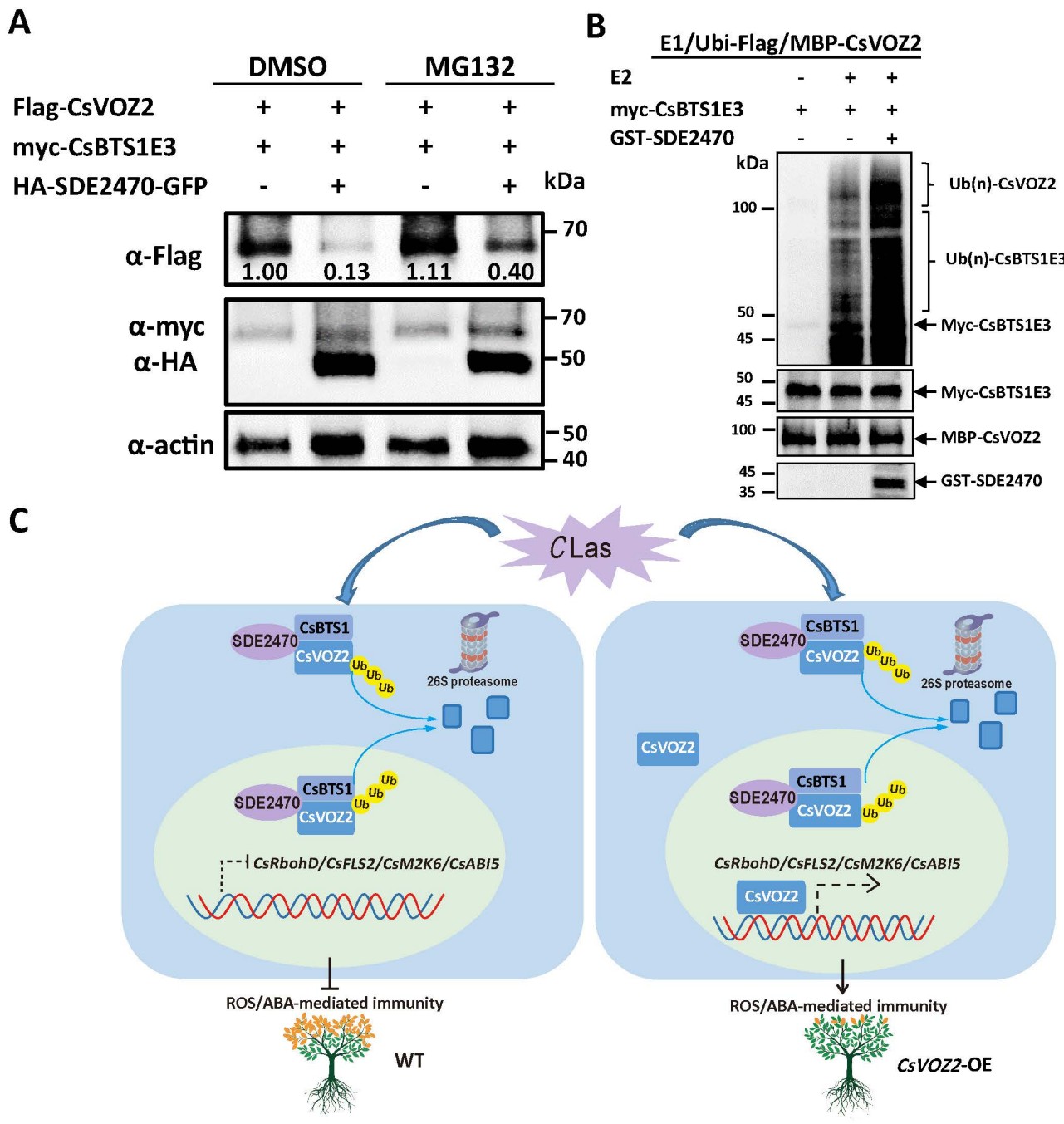

**Fig 7. SDE2470 promotes CsBTS1E3-mediated CsVOZ2 degradation via 26S proteasome pathway.** (A) The effect of SDE2470 on CsBTS1E3-medaited CsVOZ2 degradation was assayed *in vivo*. Myc-CsBTS1E3 and flag-CsVOZ2 were co-transiently expressed with or without HA-SDE2470-GFP in *N. benthamiana* leaves for WB assays. The presence of the target proteins was confirmed with corresponding tag antibodies. Actin was used as internal control. The abundance of flag-tagged CsVOZ2 protein in each lane was quantified using ImageJ and normalized against the corresponding actin abundance. Then the ratio of the relative CsVOZ2 abundance in the first lane was set to 1.00 for relative comparison. (B) The effect of SDE2470 on CsBTS1E3-mediated CsVOZ2 ubiquitination was assayed *in vitro*. Proteins were detected by corresponding tag antibodies. (C) A proposed working model: Under homeostatic conditions, CsBTS1E3 mediates the degradation of the vascular transcription factor CsVOZ2 via the 26S proteasome pathway. Upon *C*Las infection, SDE2470 enhances the E3 ligase activity of CsBTS1E3, thereby promoting CsVOZ2 degradation. The subsequent reduction in CsVOZ2 protein abundance suppresses ROS- and ABA-mediated defenses and facilitates symptom development. In contrast, overexpression of CsVOZ2 enhanced the expression of genes involved in ROS and ABA signaling pathways, thereby inhibiting *C*Las proliferation and alleviating disease symptom progression.

it remains to be determined whether citrus VOZ2 is specifically expressed in the phloem tissue, mediates phloem-based immunity, and whether this function also confers resistance to psyllid feeding.

Critical to VOZ stability manipulation is BTS, an iron-sensing hub with dual roles in immunity and development. Notably, effector AvrRps4 directly targets BTS to enhance iron uptake and *Pseudomonas syringae* colonization in Arabidopsis plants [20]. Beyond iron uptake regulation, BTS functions as an evolutionary flexible scaffold: it mediates VOZ degradation for immune fine-tuning [19], supports viral restriction in *N. benthamiana* [38]. In particular, BTSa shows vascular-specific expression in uninfected roots and is systemically induced upon rhizobial infection, and it facilitates symbiotic nitrogen fixation in legumes associated with NSP1 monoubiquitination [39]. This functional plasticity underscores why BTS mislocalizes to chloroplasts causing developmental defects and knockout is embryonically lethal [40]. BTS is a multifunctional hub that integrates iron sensing, stress adaptation, and developmental cues through ubiquitin-dependent pathways. Its functional plasticity is evidenced by its dual roles in the degradation (e.g., of iron regulators) and stabilization (e.g., of NSP1) of client proteins across species. Interestingly, in Arabidopsis, two BTS homologues (BTSL1 and BTSL2) mediate the degradation of IMA1, which leads to the loss of IMA1 from the iron-uptake complex, thereby restricting iron uptake to suppress harmful bacteria growth [41]. Although we successfully verified the function of its E3 domain, attempts to clone the full-length CsBTS1 were unsuccessful. Given this complexity, the cloning difficulty is likely attributable to the its large size (>200 kDa), multi-domain architecture (including three HHE, a CHY-type, and a RING domain) and inherent instability [20]. This instability may result from the regulatory plasticity and its rapid turnover in planta, leading to low protein abundance and a corresponding scarcity of full-length mRNA levels. Consequently, our current inability to obtain the full-length hinders investigation into whether CsBTS1 contributes to iron manipulation to support *C*Las colonization. Thus, it would be an interesting objective for future research.

In conclusion, the present study demonstrated that the *C*Las effector SDE2470 is a broad-spectrum immune suppressor. It facilitates CsBTS1-mediates degradation of CsVOZ2 via 26S proteasome pathway, thereby suppressing citrus immunity and subsequently promoting *C*Las proliferation (Fig 7C). In the phloem microenvironment where *C*Las exerts its pathogenic effects, deciphering how *C*Las manipulates phloem biology through effector-targeted transcription factors and regulatory proteins represents a critical research frontier.

## Materials and methods

### Plant materials and growth conditions

*Nicotiana benthamiana* and *Arabidopsis thaliana* (Columbia ecotype) plants were grown in controlled chambers maintained at 25±2°C with a 16-hour light/8-hour dark photoperiod and a light intensity of 12,000 lux. Wild-type (WT) and transgenic Wanjincheng orange (*Citrus sinensis*) plants were cultivated in a greenhouse under natural light conditions at 28°C. The greenhouse is located at the National Citrus Germplasm Repository in Chongqing, China.

### Bacterial strains

*Escherichia coli* DH5α and BL21 strains were utilized for plasmid cloning and *in vitro* protein expression, respectively. *Agrobacterium tumefaciens* strains GV3101, K599, and EHA105 were employed for transient expression in plants, transgenic hairy root induction and citrus plant transformation, respectively. *Xanthomonas citri* subsp. *citri* (*Xcc*) and *Candidatus* Liberibacter asiaticus (*C*Las) were used to evaluate disease resistance in transgenic citrus plants. *Pseudomonas syringae* pv. *tomato* DC3000 (*Pst* DC3000) was applied for susceptibility assays in *Arabidopsis* plants.

### Plasmid construction

All vectors and corresponding primers utilized in this study are listed in S3 Table. All constructed plasmids were verified by PCR amplification and Sanger sequencing (Tsingke Biotechnology Co., Ltd., Chongqing, China).

### *Agrobacterium*-mediated transient expression in *N. benthamiana* and *C. sinensis*

The recombinants were introduced into *A. tumefaciens* GV3101, and then vectors were transiently expressed in *N. benthamiana* or citrus leaves alone or in pairs. At 2 days post-infiltration (dpi), the infiltrated leaves were sampled from the plants and subjected to different treatments for various experimental purpose.

For ROS accumulation assay, *A. tumefaciens* cultures harboring either the PVX-GFP control vector or the PVX-SDE2470 construct were infiltrated into *N. benthamiana* plants. At 24 hours post-infiltration (hpi), *A. tumefaciens* cells carrying PVX-BAX or PVX-INF1 were further injected into the same infiltrated leaf areas. At 2 days post-infiltration (dpi), the whole leaves were detached from the plants and subjected to ethanol decolorization, followed by histochemical staining using 3,3'-diaminobenzidine (DAB) staining assays [8,42]. Reactive oxygen species (ROS), particularly hydrogen peroxide ($H_2O_2$) indicated by brown coloration in *N. benthamiana* leaves following DAB histochemical staining was quantified with ImageJ. For BiFC assay, leaves discs were subjected to fluorescence observation via confocal microscope (Olympus, FV3000). For LCA assay, whole leaves were sprayed with 1 mM D-luciferin (Yesen) dissolved in 0.01% (v/v) Triton X-100 and kept in the dark for 5 minutes before signal detection with a ChemiX imaging system. The average luciferase activity of each treatment was analyzed using ImageJ software with at least three leaves. All experiments were performed in three independent biological replicates or repeated three times, with reproducibly comparable results.

### The susceptibility assay of *SDE2470*-OE transgenic citrus plants to *Xcc*

The *SDE2470* overexpression (*SDE2470*-OE) transgenic Wanjincheng citrus plant has been evaluated to promote *C*Las infection [12]. In order to determine its effect on other citrus pathogens, five citrus leaves from *SDE2470*-OE #4 plant were inoculated with citrus canker *Xcc* at suspension (1 × 10^5 CFU/mL) via pinprick method and photographed at 15 dpi. Then the lesion size and *Xcc* biomass were assessed according to protocols [43].

### Genetic transformation of *A. thaliana* and *Pst* DC3000 inoculation assay

The coding sequences of *SDE2470* were amplified and cloned into the plant expression vector pCAMBIA-GN-35S-MCS-NOS (pLGN). Recombinant plasmids were introduced into *Agrobacterium tumefaciens* GV3101, which was then used to transform *A. thaliana* via the floral dip method [44]. Harvested seeds were surface-sterilized, vernalized on selection medium (1/2 Murashige and Skoog supplemented with 50 μg/mL kanamycin) for 2 days at 4°C in the dark, and subsequently transferred to a growth chamber under long-day conditions (16 h light/8 h darkness) at 22°C. Homozygous T3 transgenic lines were selected using 1/2 MS medium, and their identity was confirmed by RT-PCR and GUS staining.

To assess the functional roles of *SDE2470* in *A. thaliana* resistance, three T3 homozygous transgenic lines with three biological replicates per line were inoculated with *Pseudomonas syringae* pv. *tomato* DC3000 (*Pst* DC3000), with wild-type (WT) plants serving as controls. Leaf discs (6 mm diameter) were collected at 2 and 4 days post-inoculation (dpi) for bacterial biomass quantification according to established protocols [45]. For each transgenic line, at least three leaves were inoculated. Statistical analysis was performed to evaluate the significance of observed differences. Meanwhile, the phenotype was recorded at 4 dpi and the necrosis density was evaluated with ImageJ.

### Genetic transformation of *C. sinensis* and *C*Las inoculation assay

To generate transgenic citrus plants, *Agrobacterium tumefaciens* strain EHA105 harboring the recombinant plasmid pNmGFPer-Flag-CsVOZ2 was utilized for transformation of epicotyl segments from *C. sinensis* 'Wanjincheng', following established protocols [46]. The successful transformation of buds was confirmed via GFP fluorescence observation, RT-qPCR, and Western blot with anti-Flag antibody (AlpVHH). Subsequently, these transgenic buds were grafted onto two year-old citrange rootstocks and propagated for further evaluation.

For the generation of citrus hairy roots overexpressing *CsBTS1E3* or silencing *CsVOZ2*, *Agrobacterium tumefaciens* strain K599 harboring the recombinant plasmids pNmGFPer-myc-CsBTS1E3/E3M and pNmGFPer-CsVOZ2-RNAi was used to transform *Poncirus trifoliate* stem sections. In the *CsVOZ2*-RNAi construct (Fig 3C), the interfering sequence targeted the zinc-finger region within Domain-B contains a CCCH-type zinc-finger motif essential for DNA binding [29]. The transformation procedures were conducted using *Agrobacterium*-mediated methods as previously described [47]. The successful induction of hairy roots was verified through GFP fluorescence observation, PCR and RT-qPCR assays.

After generating transgenic plants and hairy roots, *C*Las-infected buds were graft-inoculated onto the plants (approximately 2 cm away from the transgenic scion) or directly onto the aerial parts of the hairy roots, respectively. Each transformation, along with wild-type (WT) and empty vector (EV) controls, was performed with at least three biological replicates. *C*Las detection was conducted via qPCR at 4 weeks post-inoculation (wpi) following a published protocol [48]. For transgenic plants, monitoring continued until 3 mpi, while hairy roots were analyzed only at the 1 mpi onetime point.

## Protein stability assay

To explore the effect of SDE2470 on CsVOZ2 protein stability, myc-tagged SDE2470 or GUS vectors was transiently co-expressed with flag-tagged CsVOZ2 in *N. benthamiana*, which were subjected to MG132 (50 μM) or the solvent DMSO treatment at 24 hpi. Two days post-inoculation, leaf samples were collected for protein extraction according to the manufacturer's protocol (Solarbio, cat# BC3720) and subsequently analyzed by SDS-PAGE. When evaluate the effect of CsBTS1E3 or CsBTS1E3M on CsVOZ2 protein stability, it was performed similarly. All experiments were conducted three times with comparable results.

## ROS and ABA contents measurement

For ROS and ABA content measurement, 3–4 fully expanded mature leaves per transgenic line were collected, ground into powder and measured independently according to the manufacturer's protocols (Sinobestbi, YX-182519P and YX-010201P). Then the ROS/ABA content levels were statistically analyzed with one-way ANOVA test.

## Protein expression and ubiquitination assay in *E. coli*

*In vitro* ubiquitination assay was carried out according to protocol [49]. Recombinant containing myc-tagged CsBTS1E3 or CsBTS1E3M, and MBP-tagged CsVOZ2 were co-transformed and expressed with or without E1/E2/Ubi-Flag in *Escherichia* coli strain BL21 (DE3). Target proteins was induced by the addition of 0.5 mM IPTG at $OD_{600}=0.4$-0.6 and at 28°C for 10–12 h. After induction, the bacteria were then stored at 4°C overnight. Then, the crude protein extracts were separated by SDS-PAGE and analyzed by Western blotting with the corresponding antibodies (AlpVHH). The experiments were performed three times yielding consistent results.

## DNA/RNA extraction, RNA-seq and quantification RT-qPCR

Plant DNA was extracted from leaves or roots with DNA extraction kit (Beyotime). *C*Las presence was assayed with OI1/OI2c by PCR and quantified via qPCR [48,50]. The Plant total RNA was extracted with RNAiso Plus (Takara). Genomic DNA removal and cDNA synthesis was conducted with All-In-One 5 × RT MasterMix (ABM). The qPCR was performed with specific primers (S3 Table) and SYBR qPCR SuperMix Plus Kit (ABM) on qTower3G real-time PCR system (Analytik Jena AG). Each assay was applied with three biological replicates and three technical replicates. The *C. sinensis actin-7* gene (XM_006464503.4) was used as internal control. The relative gene expression levels were calculated using the $2^{-\Delta\Delta Ct}$ method [51]. Graphpad Prism 9.5 were used for statistical analysis.

For RNA-seq analysis, 3–4 fully expanded tender leaves were collected from healthy *CsVOZ2*-OE#2 and WT plants. The samples were sent to BGI for RNA extraction and subsequent sequencing on the Illumina HiSeq 2500 platform

(T7 mode). Raw reads were quality-trimmed and aligned to the *Citrus sinensis* genome V3 obtained from the Citrus Pan-genome to Breeding Database (http://citrus.hzau.edu.cn/index.php). Subsequent data processing and analyses were performed as described previously [52]. Differential expressed genes (DEGs) were identified by DESeq2 (v.1.4.5) with $P$-adjust ≤ 0.05 and absolute value of $\log_2$FC ≥ 1.0. For tissue-specific expression enrichments, DEGs were mapped against *A. thaliana* in scPlantDB [53]. Then phloem expressed candidates were enriched in KEGG pathway.

## Statistical analyses

All data are presented as the mean ± SD from at least three biological replicates. Statistical significance was determined using a two-tailed Student's *t*-test for comparisons between two groups, or one/two-way ANOVA for multiple groups with GraphPad Prism 9.5. Significant differences (*$P < 0.05$, **$P < 0.01$, ***$P < 0.001$, ****$P < 0.0001$) among groups are indicated by different number of asterisks. Detailed descriptions of the statistical tests are provided in the respective figure legends.

## Supporting information

**S1 Fig. Verification of the *SDE2470* overexpression transgenic citrus and Arabidopsis plants.** (A-C) GUS staining, PCR detection and RT-qPCR verification of *SDE2470*-OE transgenic citrus plants. The relative expression level of *SDE2470* was moralized to citrus *actin 7* and statistical significance was determined using one-way ANOVA test (*$P < 0.05$ and ****$P < 0.0001$). (D-F) Kanamycin resistance selection, GUS staining and PCR verification of T3 *SDE2470-OE* transgenic *Arabidopsis thaliana* plants.
(TIF)

**S2 Fig. Phylogenetic and expression analyses of CsVOZ2.** Phylogenetic analysis of CsVOZ2 and its homologs from other plants. Relative expression of *CsVOZ2* in response to *C*Las infection in citrus plants.
(TIF)

**S3 Fig. The characteristics analyses of CsVOZ2.** (A-B) The homodimer formation of CsVOZ2 was verified by LCA and BiFC. (C) Transcriptional activation activity assay of CsVOZ2 in yeast via Y2H system. BD-CsVOZ2 and AD empty vector were co-transformed into yeast strain and cultured in SD/-Leu-Trp + X-α-gal and SD/Leu-Trp-His-Ade + X-α-gal plates (lacking leucine, tryptophan, histidine and adenine and supplemented with X-α-gal).
(TIF)

**S4 Fig. Subcellular localization of SDE2470 and CsVOZ2.** (A) Subcellular localization of SDE2470 and CsVOZ2 individually by transient co-expression with H2B-mCherry or cytoplasmic marker mCherry in *N. benthamiana* leaves. Scale bars = 20 μm. (B) Co-subcellular localization of SDE2470 and CsVOZ2 by transient co-expression with H2B-mCherry or cytoplasmic marker mCherry in *N. benthamiana* leaves. Scale bars = 20 μm.
(TIF)

**S5 Fig. Verification *CsVOZ2*-OE and RNAi transgenic citrus plants/hairy roots.** (A-C) Verification of *CsVOZ2*-OE transgenic citrus plants with PCR, WB and RT-qPCR assays. Actin served as a control. The relative expression level of *CsVOZ2* was moralized to citrus actin and statistical significance was determined using one-way ANOVA test (****$P < 0.0001$). (D-E) Verification of *CsVOZ2*-RNAi citrus hairy roots with PCR and RT-qPCR assays. Actin served as control. The relative expression level of *CsVOZ2* was moralized to citrus actin and statistical significance was determined using one-way ANOVA test (****$P < 0.0001$).
(TIF)

**S6 Fig. Transcriptomic profiling analyses of *CsVOZ2*-OE plants based on RNA-seq.** (A) Volcano plot displaying differentially expressed genes (DEGs) in healthy *CsVOZ2*-OE *vs*. WT plants. (B) Top 20 of DEGs enriched KEGG pathways

in *CsVOZ2*-OE *vs*. WT plants. (C) Visualization of DEGs involved pathways in *CsVOZ2*-OE transgenic citrus plants. (D) Tissue-specific expression enrichment of DEGs in *CsVOZ2*-OE *vs*. WT plants. (E) Functional enrichment analysis of DEGs associated with the phloem complex.
(TIF)

**S7 Fig. CsBTS1 identification and its E3 ligase activity assay.** (A and C) Transcriptional levels of *CsVOZ2* and *CsBTS1/2* genes in *SDE2470*-OE transgenic citrus plants determined by RT-qPCR. The relative expression level of *CsVOZ2* and *CsBTSs* was normalized to citrus actin and statistical significance was determined using one/two-way ANOVA (****$P < 0.0001$). (B) Heatmap of DEGs encoding E3 ligases in *SDE2470*-OE transgenic citrus plants via RNA-seq analyses. (D) Phylogenetic tree and domain architecture of CsBTSs homologues in plants. Multiple sequence alignment was performed using ClustalW. A phylogenetic tree was constructed with MEGA 11 based on 1,000 bootstrap replicates. Three CsBTS genes were retrieved from the *Citrus sinensis* V1 genome in the Citrus Pan-genome2breeding Database. For CsBTS1 (Cs2g12820), three protein isoforms were identified from the NCBI database and the one cloned (XP_006468730) in our study was used for phylogenetic construction. (E) Subcellular localization of CsBTS1E3. CsBTS1E3 (782–1242 aa) is a truncate of the full length CsBTS1 (XP_006468730) as indicated by the schematic diagram. (F) *In vitro* assay of E3 ligases activity of CsBTS1E3. Key residues critical for ligase activity were mutated (indicated in red in the schematic) to assess their functional role.
(TIF)

**S1 Table. The complete list of differentially expressed genes identified in *CsVOZ2*-OE *vs*. WT.**
(XLSX)

**S2 Table. Differentially expressed genes potentially expressed in the phloem complex in *CsVOZ2*-OE *vs*. WT.**
(XLSX)

**S3 Table. Primers were used in this study.**
(XLSX)

**S4 Table. Source data for graphs in this study.**
(XLSX)

**S1 Data. Original western blot images.**
(PDF)

## Acknowledgments

We are deeply grateful to Dongping Lv from Shanghai Jiao Tong University for providing the ubiquitination vectors, to Professor Xianchao Sun from Southwest University for supplying the pART27 vector, and to Professor Xiuping Zou from Southwest University for kindly providing the pNmGFPer vector.

## Author contributions

Conceptualization: Shimin Fu, Xuefeng Wang.

Formal analysis: Shimin Fu, Xiaofeng Yang, Zuhui Yang.

Funding acquisition: Shimin Fu, Xuefeng Wang.

Investigation: Shimin Fu, Changyong Zhou, Xuefeng Wang.

Methodology: Shimin Fu, Xiaofeng Yang, Haoqing Zhao, Jiajun Wang, Mingyue Qin.

Resources: Changyong Zhou, Xuefeng Wang.

Supervision: Shimin Fu, Changyong Zhou, Xuefeng Wang.

Writing – original draft: Shimin Fu.

Writing – review & editing: Shimin Fu, Jiajun Wang, Mingyue Qin, Changyong Zhou, Xuefeng Wang.

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
