## [Decision Letter · Decision Letter 0]

12 Oct 2025

'*Candidatus* Liberibacter asiaticus' effector SDE2470 facilitates citrus transcription factor CsVOZ2 degradation via BRUTUS E3 ligases

PLOS Pathogens

Dear Dr. Fu,

Thank you for submitting your manuscript to PLOS Pathogens. Your manuscript has been reviewed by three reviewers and two members of the Editorial Board. All reviewers recognize the novel mechanistic insights into Citrus Huanglongbing provided by the identification of the *Candidatus* Liberibacter asiaticus effector SDE2470 and the elucidation of its interactions with the host proteins CsVOZ2 and CsBTS1. They highlighted the robust and comprehensive experimental framework supporting these findings. However, they also raised several important points requiring clarification and improvement. These include the need for greater mechanistic depth linking CsVOZ2/CsBTS1 regulation to downstream defense responses and transcriptomic pathways, clarification of methodological and statistical details such as biological replicates and analysis descriptions, language and structural revisions to enhance clarity and readability, and experimental clarifications concerning the transcriptional role of CsVOZ2, the rationale for using Nicotiana benthamiana assays, and the interpretation of certain immunoblot results. After careful consideration, we feel that it has merit but does not fully meet PLOS Pathogens's publication criteria as it currently stands. Therefore, we invite you to submit a revised version of the manuscript that addresses the points raised during the review process.

Please submit your revised manuscript within 60 days Dec 11 2025 11:59PM. If you will need more time than this to complete your revisions, please reply to this message or contact the journal office at plospathogens@plos.org. Please include the following items when submitting your revised manuscript:

We look forward to receiving your revised manuscript.

Kind regards,

Sébastien Bontemps-Gallo

Academic Editor

PLOS Pathogens

Savithramma Dinesh-Kumar

Section Editor

Sumita Bhaduri-McIntosh

Editor-in-Chief

PLOS Pathogens

orcid.org/0000-0003-2946-9497

Michael Malim

PLOS Pathogens

orcid.org/0000-0002-7699-2064

**Journal Requirements:**

2) We noticed that you used the phrase 'data not shown' in the manuscript. We do not allow these references, as the PLOS data access policy requires that all data be either published with the manuscript or made available in a publicly accessible database. Please amend the supplementary material to include the referenced data or remove the references.

- TM on pages: 18, and 19.

5) We notice that your supplementary Figures are included in the manuscript file. Please remove them and upload them with the file type 'Supporting Information'. Please ensure that each Supporting Information file has a legend listed in the manuscript after the references list.

Potential Copyright Issues:

i) Please confirm (a) that you are the photographer of 1A, 1C, 1E, 3A, 3C, 5A, S1A, S1D, S1E, and S3C, or (b) provide written permission from the photographer to publish the photo(s) under our CC BY 4.0 license.

ii) Figure 7C. Please confirm whether you drew the images / clip-art within the figure panels by hand. If you did not draw the images, please provide (a) a link to the source of the images or icons and their license / terms of use; or (b) written permission from the copyright holder to publish the images or icons under our CC BY 4.0 license. Alternatively, you may replace the images with open source alternatives. See these open source resources you may use to replace images / clip-art:

7) Please amend your detailed Financial Disclosure statement. This is published with the article. It must therefore be completed in full sentences and contain the exact wording you wish to be published.

**Reviewers' Comments:**

Reviewer's Responses to Questions

**Part I - Summary**

Reviewer #1: The *Candidatus* Liberibacter asiaticus (CLas) caused Citrus Huanglongbing (HLB) is a destructive disease in citrus industry worldwide. This manuscript provides novel and valuable insights into the role of a CLas effector (SDE2470) and its interactions with interaction with host CsVOZ2 and CsBTS1, providing mechanistic insight into host–pathogen interplay.The integration of yeast two-hybrid, BiFC, co-IP, RNAi, and VIGS approaches demonstrates a strong experimental framework. Findings of this study are significant to the citrus HLB field, as they highlight how effectors manipulate host protein networks beyond defense suppression, possibly rewiring transcriptional regulation and signaling. Although the link between CsVOZ2/CsBTS1 regulation and HLB disease progression remains correlative, mechanistic depth could be improved by showing downstream signaling/transcriptome effects.

Reviewer #2: The authors investigated the role of the *Candidatus* Liberibacter asiaticus (CLas) effector SDE2470 in citrus Huanglongbing (HLB) pathogen. They demonstrated that SDE2470 physically interacts with the citrus transcription factor CsVOZ2 and promotes its proteasomal degradation via the E3 ligase CsBTS1. Overexpression of CsVOZ2 enhanced citrus resistance to CLas, while its knockdown increased susceptibility. Mechanistically, SDE2470 strengthened the CsBTS1–CsVOZ2 interaction and boosted CsBTS1’s ubiquitin ligase activity, thereby suppressing ROS- and ABA-mediated defense pathways and facilitating bacterial infection. However, the manuscript requires substantial improvement in language clarity (many grammatical errors and redundancies).

Reviewer #3: Based on previous studies identifying SDE2470 as a virulence factor of CLas, Fu et al. identified CsVOZ2 as an interactor of SDE2470 and demonstrated its role in suppressing CLas infection. Their findings revealed that CsVOZ2 was associated with plant defense mechanisms. Further investigation showed that CsVOZ2 interacted with CsBTS1, an E3 ubiquitin ligase that mediated the degradation of CsVOZ2 via the 26S proteasome pathway. The authors proposed that SDE2470 enhanced the E3 ligase activity of CsBTS1 through their interaction, thereby strengthening the CsBTS1-CsVOZ2 association and promoting CsVOZ2 degradation. This process ultimately compromised plant defense and facilitates CLas infection.

**Part II – Major Issues: Key Experiments Required for Acceptance**

Reviewer #1: Major suggestion

If possible, examined expression of well-characterized defense-related genes (e.g., PR1, NPR1-related pathways, JA/SA markers) in CsVOZ2/CsBTS1-silenced plants with or without effector expression, to directly link interactions with immune outcomes.

Reviewer #2: Major Comments

1. Many results mention qPCR, WB, RNA-seq etc., but you don’t specify the number of biological replicates in the Results section (only sometimes in Methods). These need consistency.

2. Figures are said to be supported by statistical analysis (e.g., Fig 3D, Fig 5D), but the text doesn’t always describe statistical tests or significance levels.

3. In Discussion (p.12–14), you link CsVOZ2 to phloem carbohydrate metabolism, but the evidence is indirect (based on RNA-seq prediction). Phrase more cautiously.

4. The introduction lists many effectors in a very long paragraph. Breaking into subparagraphs (effector categories, classical vs non-classical, functions) would improve readability.

5. Needs rephrasing “Therefore, SDE2470 served as a board basal immune suppressor to facilitate pathogen infection”

6. In Results, you mention failure to clone full-length CsBTS1 but used truncation (E3 domain). Discuss limitations this causes in interpretation.

7. Some results mention supplementary figures (S7B, S7D etc.) but no description in text beyond “data shown.” Expand slightly.

8. Simplify language further for non-specialist readers. Phrases like “hijacks a plant protein” are fine, but “CsVOZ2 made plants more resistant” could be rephrased more clearly.

Reviewer #3: While this story is interesting, the underlying logic and experimental validation require further reinforcement to be fully convincing.

Major comments:

1. SDE2470, CsVOZ2 and CsBTS1 interacted with each other, why did not CsBTS1 mediate SDE2470 degradation, but CsVOZ2?

2. Several assays were performed in the artificial materials, like N. benthamiana, and overexpression plants, but not in the CLas-infected plants. For example, the expression of genes, the ubiquitination test, protein-protein interaction. Lines 192-195, when RT-qPCR assay showed no change in CsVOZ2 gene expression on SDE2470-OE plants, it is illogical to test the CsVOZ2 protein level in N. benthamiana.

3. A key aspect of this story was the transcriptional activation capability of CsVOZ2; however, the evidence provided remains limited. Fig. S3A and S3B only demonstrate self-interaction of CsVOZ2, without confirming the formation of homodimers. Additionally, Fig. S3C illustrates self-interaction in a yeast two-hybrid assay but does not substantiate transcriptional activation activity in a yeast one-hybrid context. Therefore, the claim that “CsVOZ2 is a truly transcription factor as homodimers” (lines 148-150) appears overstated. Further evidence is necessary to conclusively establish the transcriptional activation function of CsVOZ2.

4. There are some flaws in Fig. 7B. What did MBP antibody detect for? Was some protein missing? Why could myc antibody detect a band without myc-CsBTS1E3 in the first lane?

**Part III – Minor Issues: Editorial and Data Presentation Modifications**

Reviewer #1: Editorial and minor suggestion

1. Line 15 "remains difficult to manage" to "lacks effective control strategies"

2. Line 57-75 Streamline the background on CLas effectors to avoid redundancy and transition more directly to the rationale for studying SDE2470. In particular, the function of SED5 (same as SDE2470 in this study) have been previously reported in Zhao et al 2025 (doi: 10.1126/science.adq7203). It is suggested to add more detial of previous work related to SDE5 or SDE2470 in this section.

3. Line 149 "harbors transcriptional activation activity" to "exhibits transcriptional activation activity"

4. Line 165 "...was significantly lower in CsVOZ2-OE than that in WT plants..." to "...was significantly lower in CsVOZ2-OE plants than in WT plants..."

5. Line 166 "whereas CLas colonizes much faster..." to "CLas colonized more rapidly"

6. Line 243 "...enhances CsBTS1E3 degradation on CsVOZ2..." to "enhances CsBTS1E3-mediated degradation of CsVOZ2"

7. Line 311 "deciphering how this bacterium manipulates..." to "deciphering how CLas manipulates..."

8. The legends of some figures can be improved. Such as explicitly state whether data are from citrus or N. benthamiana, number of replicates, type of statistical test used.

Reviewer #2: Minor Comments

1. “These studies have elucidated the molecular mechanisms. SDE1…” to combine into one sentence or rephrase for flow.

2. Typo: “share 55~61% amino acid sequence similarity” to “shares 55–61%” (and use en dash).

3. Acronyms: define ROS, ABA, UPS at first mention in Introduction, not later.

4. “were tested on SDE2470-overexpressing (OE) transgenic Arabidopsis and citrus plants” to rephrase as “were tested in SDE2470-overexpressing…”

5. Sentence: “Although CLas was detected in both WT and CsVOZ2-OE plants…” better as “CLas was detected in both, but its population was significantly lower in CsVOZ2-OE plants.”

6. Inconsistency: “26 proteasome pathway” vs. “26S proteasome pathway.” Use “26S” consistently.

7. Replace “in negative regulation of host immunity” to “negatively regulates host immunity.”

Reviewer #3: Minor comments:

1. Why were the positions of 358 and 361 in CsBTS1E3M mutated?

2. The loading of reference proteins was different in this manuscript, such as Fig. 4C, 4B, and 7A. Fig. 7B.

PLOS authors have the option to publish the peer review history of their article (what does this mean? ). If published, this will include your full peer review and any attached files.

**Do you want your identity to be public for this peer review?** For information about this choice, including consent withdrawal, please see our Privacy Policy .

Reviewer #1: No

Reviewer #2: **Yes: ** DK Ghosh

Reviewer #3: No

**Figure resubmission:**

**Reproducibility:**



---

## [Decision Letter · Decision Letter 1]

10 Dec 2025

Dear Associate Prof. Fu,

We are pleased to inform you that your manuscript ''*Candidatus* Liberibacter asiaticus' effector SDE2470 facilitates citrus transcription factor CsVOZ2 degradation via BRUTUS E3 ligases' has been provisionally accepted for publication in PLOS Pathogens.

Best regards,

Sébastien Bontemps-Gallo

Academic Editor

PLOS Pathogens

Shou-Wei Ding

Section Editor

PLOS Pathogens

Sumita Bhaduri-McIntosh

Editor-in-Chief

PLOS Pathogens

orcid.org/0000-0003-2946-9497

Michael Malim

Editor-in-Chief

PLOS Pathogens

orcid.org/0000-0002-7699-2064

Reviewer Comments (if any, and for reference):

Reviewer's Responses to Questions

**Part I - Summary**

Reviewer #1: The revised manuscript has addressed all the requested revisions thoroughly and appropriately. I therefore agree to accept the manuscript in its current version for publication.

Reviewer #3: The authors have addressed many of the previous comments satisfactorily, and the current version of the manuscript shows clear improvement. However, there are a few points that could be further refined. Below are specific suggestions and comments for this revised version.

**Part II – Major Issues: Key Experiments Required for Acceptance**

Reviewer #1: (No Response)

Reviewer #3: (No Response)

**Part III – Minor Issues: Editorial and Data Presentation Modifications**

Reviewer #1: (No Response)

Reviewer #3: 1. Fig. 4, the Ubi-Flag should be shown in lanes 1, 2, 4, 5 and 6.

2. The observed self-interaction of CsVOZ2 indicates the formation of homomultimers, though it does not exclusively confirm a homodimeric structure. In the Fig. S3A and B, the self-interaction of CsVOZ2 was shown, but the homodimer was not confirmed. It is therefore recommended that the authors discuss these results in the manuscript.

PLOS authors have the option to publish the peer review history of their article (what does this mean? ). If published, this will include your full peer review and any attached files.

**Do you want your identity to be public for this peer review?** For information about this choice, including consent withdrawal, please see our Privacy Policy .

Reviewer #1: No

Reviewer #3: No

---

## [Editor Report · Acceptance letter]

Dear Associate Prof. Fu,

We are delighted to inform you that your manuscript, "'*Candidatus* Liberibacter asiaticus' effector SDE2470 facilitates citrus transcription factor CsVOZ2 degradation via BRUTUS E3 ligases," has been formally accepted for publication in PLOS Pathogens.

Best regards,

Sumita Bhaduri-McIntosh

Editor-in-Chief

PLOS Pathogens

orcid.org/0000-0003-2946-9497

Michael Malim

Editor-in-Chief

PLOS Pathogens

orcid.org/0000-0002-7699-2064